# Outlier-Robust Surrogate Modeling of Ion–Solid Interaction Simulations [note 1]

**DOI:** 10.3390/e25040685

**Published:** 2023-04-19

**Authors:** Roland Preuss, Udo von Toussaint

**Affiliations:** Max-Planck-Institut für Plasmaphysik, 85748 Garching, Germany; udt@ipp.mpg.de

**Keywords:** Gaussian process, Student-t process, Bayesian optimization, plasma–wall interaction simulation, mixture likelihood

## Abstract

Data for complex plasma–wall interactions require long-running and expensive computer simulations. Furthermore, the number of input parameters is large, which results in low coverage of the (physical) parameter space. Unpredictable occasions of outliers create a need to conduct the exploration of this multi-dimensional space using robust analysis tools. We restate the Gaussian process (GP) method as a Bayesian adaptive exploration method for establishing surrogate surfaces in the variables of interest. On this basis, we expand the analysis by the Student-t process (TP) method in order to improve the robustness of the result with respect to outliers. The most obvious difference between both methods shows up in the marginal likelihood for the hyperparameters of the covariance function, where the TP method features a broader marginal probability distribution in the presence of outliers. Eventually, we provide first investigations, with a mixture likelihood of two Gaussians within a Gaussian process ansatz for describing either outlier or non-outlier behavior. The parameters of the two Gaussians are set such that the mixture likelihood resembles the shape of a Student-t likelihood.

## 1. Introduction

Simulations of particles from fusion plasma escaping confinement and interacting with the vessel wall are extremely costly in terms of computer power and time. Consequently, results from ion–solid interaction simulations, e.g., sputter rates from the software EIRENE/FZ Jülich [1], lack real-time ability and fail to provide the fast numerical access needed, e.g., by gradient-based methods traveling through multi-dimensional parameter space while searching for extremal structures. With already-acquired data as a starting basis, the method of surrogate modeling provides fast and easy access for numerical optimization methods. In the present case, the shape of utility functions used for the selection of the next optimal point [2] is relatively benign. In situations where this is not the case, the detrimental effect of spurious peaks in the utility function can partly be avoided using modified acquisition strategies [3]. The EIRENE program employs at its heart a Monte Carlo method, by which it may be assumed to produce results with uncertainty margins that follow a Gaussian distribution. However, the code itself involves tables of source rates for particles, energies and momentum, which may introduce some nonlinear behavior, at least to the variance of the results.

It has been known for a long time that a Student-t distribution offers the possibility of making the analysis more robust with respect to outliers [4,5]. In this paper, we follow this trail and investigate the Student-t process method as a surrogate surface emulator in competition with the Gaussian process method [6]. Introduced by Rasmussen et al. in Chapter 9.9 of his landmark publication “Gaussian Processes for Machine Learning” [6], the derivation and application of a Student-t process as a surrogate emulator was examined many times. Already, Yu et al. in 2007 [7] placed the TP method on a solid foundation with correct data error handling, while Shah et al. [8] approached the same marginal likelihood by integrating an inverse Wishart process prior over the covariance kernel of the Gaussian process.

In order to investigate the differences between the GP and TP method, we set up artificial test cases in one and two dimensions. The problem we want to tackle for the sputter rates caused by fusion plasma takes place in a four-dimensional physics parameter set, so we have to transfer the results of the test cases derived with artificial data to analysis of real-world data. As a side effect, the changes to the program for adaptation to the TP method are validated by our well-established algorithm emulating surrogate surfaces. To complete these investigations, we present results for fusion plasma sputter rates in a two-dimensional subspace of a four-dimensional parameter space.

Coming to real data, the situation we face in the experiment is that outliers emerge from sensor errors or instabilities in the measurement conditions, while the major part of the data is of Gaussian nature. Therefore, we want to complete this paper by a study considering a mixture likelihood of two Gaussians within the realm of the Gaussian process method [9,10]. While one Gaussian of the mixture likelihood shall cover the normally distributed data, the other Gaussian equipped with larger standard deviation is aimed at the description of the outliers. Unfortunately, the numerical analysis becomes very costly for already decent numbers of data, which is obviously the reason that studies in the literature invoked approximation methods. This should be easily understood, considering that the number of terms explodes by a factor of two to the power of the data number. From a naive point of view, this quickly seems to become intractable, entering data pools of much more than a handful of data. However, we found an intuitive approach reducing the numerical efforts to a minimum by employing Gray code, while still taking into account all terms in the evidence integral for an analytically exact result. Gray code generates a sequence of binary representations, which differs from one to the next only by one bit. This is not the case when counting bit-wise because the binary representation for, e.g., three is 011 and four is 100, so by moving from one representation to the next, three positions have to change their digit. On the contrary, with Gray code, it is possible to cover all $2^{N}$ possibilities for *N* digits, changing only a single digit between neighboring representations of the sequence (see chapter 20.2 of [11]). Applied to changes in a matrix, this enables fast computable rank-one updates, especially if, otherwise, one has to perform a complete matrix inversion. We present, first, the results stating proof of principle for this new approach and compare it to the GP/TP methods shown in the first section of this paper.

## 2. Gaussian Process Method

The problem of predicting function values in a multi-dimensional space supported by given data is a regression problem for a non-trivial function of an unknown shape. The matrix $X=(x_{1},x_{2},\ldots ,x_{N})$ consisting of *N* input data vectors $x_{i}$ of dimension $N_{\dim}$ is given. The target data $y={\left(y_{1},\ldots ,y_{N}\right)}^{T}$ is blurred by Gaussian noise of variance $\Delta_{ij}={\sigma_{d}}_{i}^{2}\delta_{ij}$. The quantity of interest is the target value $f_{*}$ at test input vector $x_{*}$ and is generated by a function f(x), which shall satisfy y=f(x)+ϵ, with 〈ϵ〉=0 and $\langle \epsilon^{2}\rangle =\sigma_{d}^{2}$i, where the brackets 〈…〉 indicate an expectation value. As a statistical process, it is fully defined by its covariance function, which is the place where we incorporate all the properties that we would like our (hidden) problem-describing function to have. For the functional form of the covariance, we chose a Gaussian-type exponent with the negative squared value of the distance between two input data vectors $x_{p}$ and $x_{q}$.
(1)$$k\left(x_{p},x_{q}\right)=\sigma_{f}^{2}\exp\left\{-\frac{1}{2}{\left|\frac{x_{p}-x_{q}}{\lambda }\right|}^{2}\right\}.$$

The neighborhood of the two data vectors should be of relevance for the smoothness of the result, which is mimicked by a length scale λ in the denominator to represent the long-range dependence of the two vectors. Moreover, since the Gaussian process method defines a distribution over functions, the width of this distribution will have some influence on our result as well. This shall be comprised by the signal variance $\sigma_{f}^{2}$. An element of the covariance matrix of the input data is abbreviated as $K_{ij}\left(\lambda ,\sigma_{f}\right)=k\left(x_{i},x_{j}\right)$, and the vector of covariances between the test input vector and a single input data is ${\left(k_{*}\right)}_{i}=k\left(x_{*},x_{i}\right)$. Finally, in addition to the above estimation of the variance of a distinct data point with $\sigma_{d_{i}}^{2}$, provided, e.g., by the EIRENE MC-simulations, we consider an overall noise in the data by a variance $\sigma_{n}^{2}$. Starting with no further information about the hyperparameters, we assume Gaussian priors with N(1, 1).

Summing up the analysis from previous papers [6,12], the probability distribution for a single function value $f_{*}$ at test input $x_{*}$ is
(2)$$p\left(f_{*}|X,y,x_{*}\right)\propto N\left({\bar{f}}_{*},\mathrm{var}{}^{\mathrm{GP}}\left(f_{*}\right)\right),$$
with mean
(3)$${\bar{f}}_{*}=k_{*}^{T}{\left(K\left(\lambda ,\sigma_{f}\right)+\sigma_{n}^{2}\Delta \right)}^{-1}y,$$
and variance
(4)$${\mathrm{var}}^{\mathrm{GP}}\left(f_{*}\right)=k\left(x_{*},x_{*}\right)-k_{*}^{T}{\left(K\left(\lambda ,\sigma_{f}\right)+\sigma_{n}^{2}\Delta \right)}^{-1}k_{*}.$$

The hyperparameters $\theta^{T}=\left(\lambda ,\sigma_{f},\sigma_{n}\right)$ determine the result of the Gaussian process method. Since we do not know a priori which setting is useful, we marginalize over them numerically by employing the marginal likelihood
(5)$$\log p^{\mathrm{GP}}\left(y|\theta \right)=\mathrm{const}-\frac{1}{2}y^{T}{\left[K\left(\lambda ,\sigma_{f}\right)+\sigma_{n}^{2}\Delta \right]}^{-1}y-\frac{1}{2}\log\left|K\left(\lambda ,\sigma_{f}\right)+\sigma_{n}^{2}\Delta \right|.$$

## 3. Student-t Process Method

With the formulae from the above section at hand, it is easy to reformulate the analysis for the Student-t Process method, where we strictly follow the papers of Yu [7] and Shah [8]. The marginal likelihood reads
(6)$$\log p^{\mathrm{TP}}\left(y|\nu ,\theta \right)∼-\frac{\nu +N}{2}\log\left\{1+\frac{y^{T}{\left[K\left(\lambda ,\sigma_{f}\right)+\sigma_{n}^{2}\Delta \right]}^{-1}y}{\nu -2}\right\}-\frac{1}{2}\log\left|K\left(\lambda ,\sigma_{f}\right)+\sigma_{n}^{2}\Delta \right|.$$

In the following, we choose ν = 3 to resemble Cauchy distributions.

While the mean of a test function value remains the same as in Equation (Equation 3), the variance becomes
(7)$${\mathrm{var}}^{\mathrm{TP}}\left(f_{*}\right)=\frac{1+y^{T}{\left[K\left(\lambda ,\sigma_{f}\right)+\sigma_{n}^{2}\Delta \right]}^{-1}y}{1+N}\cdot {\mathrm{var}}^{\mathrm{GP}}\left(f_{*}\right).$$

Here, the most important difference to the Gaussian process shows up, i.e., the dependence of the variance on the target data. It may be regarded as a crucial disadvantage of the GP method that its results are based on the input mesh only, so the outcome depends on the experimentalist’s setup of the input parameters, e.g., at which locations in space the measurements will be taken. On the other hand, the Student-t process also involves the measurement results, which ultimately provide the capability of this data analysis method to ignore outliers.

## 4. One- and Two-Dimensional Test Cases

We start with a one-dimensional test case by mapping the first N=20 Sobol data as the input to a range [−1, 1] on the *x*-axis and use a sin-model with two full periods for this range to generate the respective target data. The input was chosen to be drawn from Sobol data [13,14] in order to provide a quasi-random sample, which is space-filling on a given region of interest. Uncertainty is introduced by adding Gaussian noise, with standard deviation $\sigma_{d}$ = 0.2. In order to guarantee comparability of the results, especially with those of the Section 6, all calculations were performed for the same data set. Therefore, minor differences may show up in comparison with our previous paper published in the Proceedings of the 41st International Workshop on Bayesian Inference and Maximum Entropy Methods in Science and Engineering, Paris, France, 18–22 July 2022 [15].

Figure 1 shows the results with the GP method and the TP method on the left and right panels, respectively. In the absence of outliers, both methods give the same answer in Figure 1a,b—only the uncertainty ranges (outer green or black lines) show differences, i.e., the GP method is trying to cover all data within a broader range. However, with two outliers at hand (two data points were raised by just multiplying with a factor of three), the surrogate from the GP method (see Figure 1c) tries to follow each target value slavishly, which results in a smaller hyperparameter λ, equivalent to a bumpier behavior. For the TP method (see Figure 1d), the bumps become less pronounced for the expectation values of the surrogate surface 〈f(θ)〉 (black line) and disappear completely by just asking for the surrogate surface, obtained by inserting the expectation values of the hyperparameters f(〈θ〉) (green line), which clearly follow a sin-function. It is informative to have a look at the marginal likelihood for the hyperparameters θ. Since there are three hyperparameters, we employ two two-dimensional plots for (λ, $\sigma_{n}$) in Figure 1e,f and (λ, $\sigma_{f}$) in Figure 1g,h, where the respectively lacking third hyperparameter $\sigma_{f}$/$\sigma_{n}$ for the first/second plot is kept constant in terms of its expectation value from integration over the marginal likelihood Equations (Equation 5) and (Equation 6), respectively. The most important differences are seen for (λ, $\sigma_{f}$), i.e., Figure 1g,h. In comparison with the GP case, for λ values around 0.05, the Student-t result shows a broader structure in $\sigma_{f}$, and for $\sigma_{f}$ around 0.5, an additional structure that comprises λ-values between [0.10, 0.25]. The contributions in the marginal likelihood for this broad bump attributed to the larger λ-values between [0.10, 0.25] are responsible for the smooth functional behavior.

In order to examine these findings more thoroughly, in Figure 2, we focus on two settings of the hyperparameters deduced from the extremal structures in Figure 1h of the Student-t process. In the left panel, starting with Figure 2a for λ = 0.05, $\sigma_{f}$ = 1.5, $\sigma_{n}$ = 1, a strong obedience to the target data is enforced. Therefore, the surfaces of the marginal likelihood, computed with either $\sigma_{f}$ = 1.5 (Figure 2c) or $\sigma_{n}$ = 1 (Figure 2e), become pinned down to a relatively small λ-variation. The situation changes in the right panel with λ = 0.18, $\sigma_{f}$ = 0.7, $\sigma_{n}$ = 2.6, where we obtain broad structures for λs around 0.2, in connection with a somewhat more relaxed functional behavior in Figure 2b.

From the above, it is clear that a MAP solution would fail completely in the presence of outliers because such an approach would focus on the maximum of the probability distribution at max λ = 0.051 and max $\sigma_{f}$ = 1.61, thereby disregarding all contributions from the PDF for larger λ, along with smoother surrogates. Consequently, only the full exploitation of the marginal likelihood Equation (Equation 6) empowers the result to resemble the sin-function.

Next, we compare GP vs. TP in two dimensions (see Figure 3). A total of N=40 target data are generated by the above double period sin-function just by expanding the x-dependence to ***x*** = ${\left(x_{1},x_{2}\right)}^{T}$. Without outliers, the resulting surrogate surface (Figure 3a) is the same for GP and TP, revealing a unimodal structure in hyperparameter space (Figure 3b), along with well-defined expectation values with more or less concise variances, 〈λ〉 = 0.3 ± 0.04, $\langle \sigma_{f}\rangle$ = 1.3 ± 0.3, $\langle \sigma_{n}\rangle$ = 0.7 ± 0.4. It is certain that the MAP approach would come to the same result for the surrogate surface.

The situation changes with outliers ($N_{\mathrm{outlier}}$ = 4). The GP surrogate (Figure 3c) fails completely and features a bump in the marginal likelihood (Figure 3c), which is confined around small λ-values below 0.1 and $\sigma_{f}∼1.4$. Compared with this, the TP surrogate in Figure 3e resembles the sin-model function, where the unimodal structure in the marginal likelihood widens (see Figure 3f), as already seen in the one-dimensional case.

## 5. Results for Ion–Solid Interaction Simulations

Finally, we employ the data-analyzing tools characterized above to sputter rates generated by the ion–solid interaction simulations in a fusion plasma with EIRENE software [1]. To simulate these data, a total of 14 physics parameters are to be set on input. The most important parameters are those regarding electron density *n* and electron temperature *T*, both at two locations within the plasma, i.e., plasma center {$n_{0}$,$T_{0}$} and at the so-called pedestal {$n_{\mathrm{ped}}$,$T_{\mathrm{ped}}$} located at the plasma edge next to the separatix (last magnetic field line closed within the vessel). To begin with, we set up a test case with N=3 × 3 × 3 × 3 = 81 EIRENE sputter rate data as a function of these four parameters {$T_{0},T_{\mathrm{ped}},n_{0},n_{\mathrm{ped}}$} (results shown in Figure 4a).

In order to improve this apparently not very informative result on only a $3^{4}$ grid, we calculate the GP surrogate on a $5^{4}$-grid and take the $3^{4}$ data, being the worst in terms of variance, feed them back to EIRENE and take the resulting second $N_{2}$ = 81 data set (containing 11 doublets from initial one). This results in the initial one adding up to a total of $N_{\mathrm{tot}}=151$ data points. One can think of this as an iterative step, keeping the computation effort of the costly EIRENE runs low. The surrogate surfaces for the initial data set with N=81 EIRENE data (blue mesh) and the full data set with $N_{\mathrm{tot}}$ = 151 (red mesh) are shown in Figure 4b, with the errorbars for the same nine data points as in Figure 4a. As can be seen, the iterative step reduces the uncertainty in the target by a factor of 3.6 (and misfit by factor of three). Moreover, while the surrogate surface (blue mesh) based on initial N=81 EIRENE data shows only a maximal structure at $T_{0}=3$ keV smeared out around $n_{0}=1.26\times 10^{14}$/cm^3^, the TP surrogate surface (red mesh) has a clear maximum at $T_{0}=3$ keV and $n_{0}=1.20\times 10^{14}$/cm^3^. The lower panel of Figure 4 shows the marginal likelihood surfaces for the hyperparameters λ, $\sigma_{f}$ for the results with N=151 data. Since the TP method (Figure 4d) shows a broader shape compared to the GP method (Figure 4c), it may be inferred from the chapters above that the four-dimensional parameter space contains the results for the sputter rates, which do not fully obey a normally distributed uncertainty.

## 6. Gaussian Process Method with Mixture of Two Gaussians

The GP method as well as the TP method above try to describe all data with a unique density function. For the majority of experimental data originating from the deterministic (though sometimes unknown) physics under observation, this works out fine, with the TP method beneficially showing some robustness against outliers. In this final section, we want to follow a different approach, stating that all data are normally distributed but split into two sets with respective standard deviations. While the data in the first set are considered to originate from measurement observations with a first standard deviation $\sigma_{d_{i}}$ residing on the measurement uncertainty provided by the experimentalist (e.g., by knowing the uncertainty of the sensors), the second set is assigned to outliers. We assume the outliers to be still but poorly connected to the physics the measurement observation is targeted on. This removed relationship with the proper first data set shall be described by a second much larger standard deviation $\sigma_{{\mathrm{outlier}}_{i}}$. Consequently, it is allowed to employ the same (Gaussian) likelihood function, i.e., same mean, for both data sets and keep records for which standard deviation is applied for which data point. Since an analytic solution very quickly becomes very costly (the integral terms are the power of two, where the exponent is the number of data), most—not to say all—approaches in the literature [9,10,16,17,18] invoke in one way or another some approximation. On the contrary, we proceed with the full integral and manage calculations of data sets of order *N*=20 with a standard PC by employing Gray code (see, e.g., [11]).

Revisiting the integral for the marginal likelihood
(8)$$p\left(y|\theta ,X\right)=\int df\;p\left(y|f,\sigma_{n}\right)p\left(f|\lambda ,\;\sigma_{f},X\right),$$
we assign a mixture of two Gaussians to the likelihood term
(9)p(y|f,σn)=∏i=1Np(yi|fi,σn)=∏i=1NCdata2πσnσdiexp−12yi−fiσnσdi2∏i=1N+Coutlier2πσnσoutlieriexp−12yi−fiσnσoutlieri2,
while the Gaussian process is (still) defined by
(10)$$p\left(f|\lambda ,\;\sigma_{f},X\right)=\frac{1}{{\left(2\pi \right)}^{\frac{N}{2}}\sqrt{\det K}}\exp\left\{-\frac{1}{2}f^{T}K^{-1}f\right\}.$$
In light of the robustness skills of the Student-t process in the previous section, the normalization constants $C_{\mathrm{data}}$ and $C_{\mathrm{outlier}}$ shall be determined by requiring each mixture to resemble a Cauchy distribution,
(11)$$p_{\mathrm{Cauchy}}\left(y_{i}|f_{i},\sigma_{n}\right)=\frac{1}{\pi \sigma_{n}\sigma_{d_{i}}}{\left\{1+{\left[\frac{y_{i}-f_{i}}{\sigma_{n}\sigma_{d_{i}}}\right]}^{2}\right\}}^{-1}.$$
The resemblance shall show up for the amplitude at $y_{i}=f_{i}$ in both Equations (Equation 9) and (Equation 11),
(12)$$\frac{1}{\pi \sigma_{n}\sigma_{d_{i}}}=\frac{C_{\mathrm{data}}}{\sqrt{2\pi }\sigma_{n}\sigma_{d_{i}}}+\frac{C_{\mathrm{outlier}}}{\sqrt{2\pi }\sigma_{n}\sigma_{{\mathrm{outlier}}_{i}}}.$$
while the slower decay with distance to $f_{i}$ shall be reflected by requiring the same functional values of Equations (Equation 9) and (Equation 11) at $|y_{i}-f_{i}|=10\sigma_{n}\sigma_{d_{i}}$,
(13)$$\frac{1}{\pi \sigma_{n}\sigma_{d_{i}}}{\left\{1+{\left[10\right]}^{2}\right\}}^{-1}=\frac{C_{\mathrm{outlier}}}{\sqrt{2\pi }\sigma_{n}\sigma_{{\mathrm{outlier}}_{i}}}\exp\left\{-\frac{1}{2}{\left[\frac{10\sigma_{n}\sigma_{d_{i}}}{\sigma_{n}\sigma_{{\mathrm{outlier}}_{i}}}\right]}^{2}\right\},$$
where we drop the term with $C_{\mathrm{data}}$ of Equation (Equation 9) for being negligible against the other terms. We cross out the hyperparameter $\sigma_{n}$, employ the normalization condition in Equation (Equation 9) to obtain $C_{\mathrm{data}}$+$C_{\mathrm{outlier}}$ = 1 and obtain an iterative to-be-solved equation for the dependence of $\sigma_{{\mathrm{outlier}}_{i}}$ on $\sigma_{d_{i}}$. It turns out that for our one-dimensional *N* = 20 toy data set with a standard deviation of $\sigma_{d}$ = 0.2, the outlier standard deviation would be $\sigma_{{\mathrm{outlier}}_{i}}$ = 4.7, i.e., the artificially chosen two outliers (see Figure 1c) are well within scope. Accordingly, the normalization constants are $C_{\mathrm{data}}$ = 0.79 and $C_{\mathrm{outlier}}$ = 0.21.

With the product over the mixture, the integral in Equation (Equation 8) contains $2^{N}$ terms, which have to be summed up for obtaining the marginal likelihood. Each term is a product of the two Gaussians of the mixture (only different in their standard deviations), with the prior function Equation (Equation 10) being itself a Gaussian. Since a product of Gaussians gives again a Gaussian, this can be integrated to obtain (see Equation (A.7) in [6])
(14)$$p^{\mathrm{GP}2G}\left(y|\theta ,X\right)=\sum_{r}{\left(C_{\mathrm{data}}\right)}^{N_{\mathrm{data}}\left(r\right)}{\left(C_{\mathrm{outlier}}\right)}^{N_{\mathrm{outlier}}\left(r\right)}\frac{\exp\left\{-\frac{1}{2}y^{T}{\left[K\left(\lambda ,\sigma_{f}\right)+\sigma_{n}^{2}\Delta \left(r\right)\right]}^{-1}y\right\}}{{\left(2\pi \right)}^{\frac{N}{2}}\sqrt{\det\left[K\left(\lambda ,\sigma_{f}\right)+\sigma_{n}^{2}\Delta \left(r\right)\right]}}.$$
with ***r*** as the $2^{N}$ terms of the mixture products. Each term implies a certain number $N_{\mathrm{data}}\left(r\right)$ for “normal” data and $N_{\mathrm{outlier}}\left(r\right)$ for the outliers. Apart from that, the only difference between the terms is implanted in the matrix Δ(r) with either $\sigma_{d_{i}}$ or $\sigma_{{\mathrm{outlier}}_{j}}$ as entries on the diagonal. While the calculation of Equation (Equation 14) involves matrix inversion as the most time consuming part, by invoking Gray code, it is possible to establish a sequence of the $2^{N}$ terms in such a way that term by term, only a single element in the matrix changes, and therefore, successive rank-one updates on an initially calculated matrix inverse are sufficient for completing the summation.

In Figure 5a—like for the previous methods above— the mixture approach reproduces the model sin-function for the N=20 data very well in the absence of outliers. However, most impressive is the result for the expectation values of the surrogate in Figure 5b in the presence of two outliers, which follows nearly exactly the course of the undistracted data. Only the respective broadening of the uncertainty range gives reference to the outliers. Drawn in green is the surrogate model obtained for just inserting the expectation values of the hyperparameters to the mean of Equation (Equation 3). While in the previous case for the TP process above this policy was the most promising one to obtain the best result for the time being, now it is revealed that only the full calculation of the surrogate expectation values can unveil all peculiarities in the probability distribution function of Equation (Equation 8). The better description of the mixture approach for data containing outliers shows up in the plainly unimodal hyperparameter surfaces in Figure 5c,d as well, which is assisted by showing only a linear relationship between the hyperparameters (some weak nonlinearity for $\lambda /\sigma_{f}$). Eventually, it can be stated with ease that the applied MCMC procedure will work out fine for such type of sampling distributions and therefore needs fewer sampling steps compared to the GP/TP methods.

## 7. Conclusions

Exploring surrogate surfaces in multi-dimensional spaces has been proven to be employed advantageously by the Gaussian process (GP) method. For experimental data suffering from outliers, it is also known that the marginal posterior distribution can be made robust by acquiring, e.g., the Cauchy function instead of deferring to the Gaussian form. As shown in this paper, utilizing the Student-t process (TP) method can be performed by only a few and simple changes to an already well-established implementation of a GP algorithm. The most important difference between both methods shows up in the marginal likelihood for the hyperparameters of the covariance function, which, in the presence of outliers, becomes broader in the TP case compared to GP. The Bayesian method is to explore hyperparameter space by marginalization and let the data decide regarding the posterior probability distribution. However, with the basic assumption of normally distributed data, the GP method slavishly follows each data point within its variance, thereby generating a surrogate surface that irredeemably deteriorates in the presence of outliers. In a real-world situation with occasionally faulty measurements, the TP method offers the possibility of ignoring heavily distorted data by featuring a broader marginal probability distribution. Moreover, the TP method improves the overall result for surrogate surfaces in comparison with Gaussian processes and adds robustness with respect to outliers. However, the best results for the surrogate surfaces are obtained by a mixture Gaussian likelihood within the Gaussian process method. Although there is seemingly enormous numerical effort that one has to take for calculating $2^{N}$ terms—each involving a matrix inversion—we could present a manageable procedure by featuring Gray code to condense the inversion expenditure down to rank-one updates on the matrix under consideration. The speed up with the Gray code allows one to tackle data sets of order O(20) already on standard PCs. This certainly can be pushed further up to set numbers of ∼O(30) by applying parallelization techniques on modern HPC systems. Coming from the other end, one can think of elaborated methods (e.g., by splitting [19]), which lend a hand in downsizing larger data pools to a size within range of our mixture approach. 

## Figures and Tables

**Figure 1 entropy-25-00685-f001:**
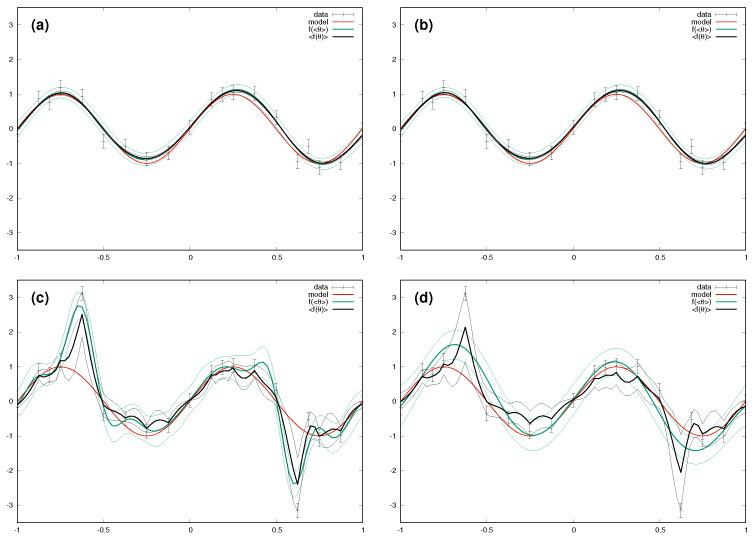
N=20 data points. Left panel (**a**,**c**,**e**,**g**) Gaussian process (GP). Right panel (**b**,**d**,**f**,**h**) Student-t process (TP). (**a**,**b**) Normally distributed data following a sin-model. (**c**,**d**) Normally distributed data following a sin-model, but the fifth and fifteenth data point were multiplied by a factor of three to simulate outliers. (**e**,**g**) GP hyperparameter surfaces for data with outliers, 〈λ〉 = 0.1 ± 0.2, $\langle \sigma_{f}\rangle$ = 1.2 ± 0.3, $\langle \sigma_{n}\rangle$ = 2.1 ± 1.2; (**f**,**h**) TP hyperparameter surfaces for data with outliers, 〈λ〉 = 0.3 ± 0.6, $\langle \sigma_{f}\rangle$ = 1.2 ± 0.7, $\langle \sigma_{n}\rangle$ = 1.9 ± 1.0.

**Figure 2 entropy-25-00685-f002:**
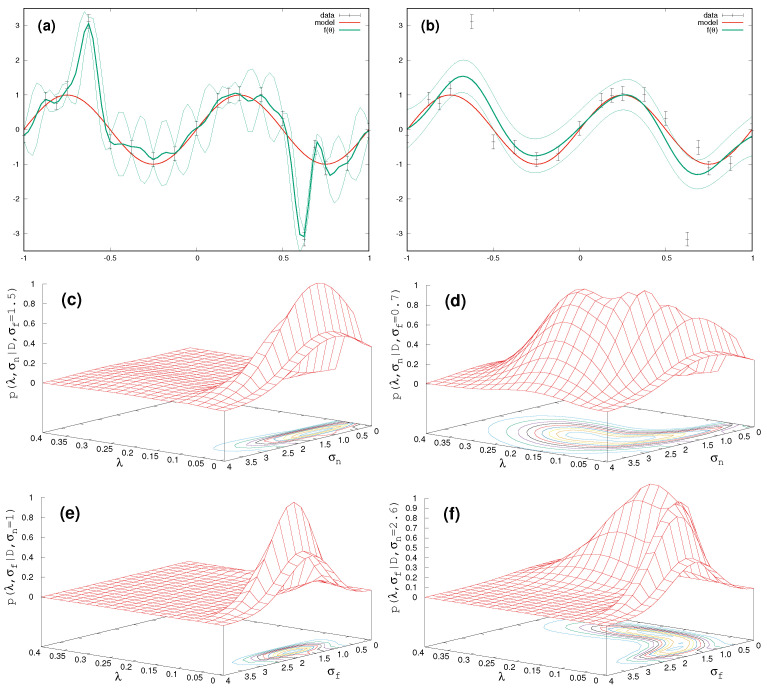
Surrogate model from Student-t process for N=20 data points, with two outliers for two settings of the hyperparameters in the extremal structures of Figure 1h. (**a**) λ = 0.05, $\sigma_{f}$ = 1.5, $\sigma_{n}$ = 1 with respective hyperparameter surfaces (**c**,**e**). (**b**) λ = 0.18, $\sigma_{f}$ = 0.7, $\sigma_{n}$ = 2.6 with respective hyperparameter surfaces (**d**,**f**).

**Figure 3 entropy-25-00685-f003:**
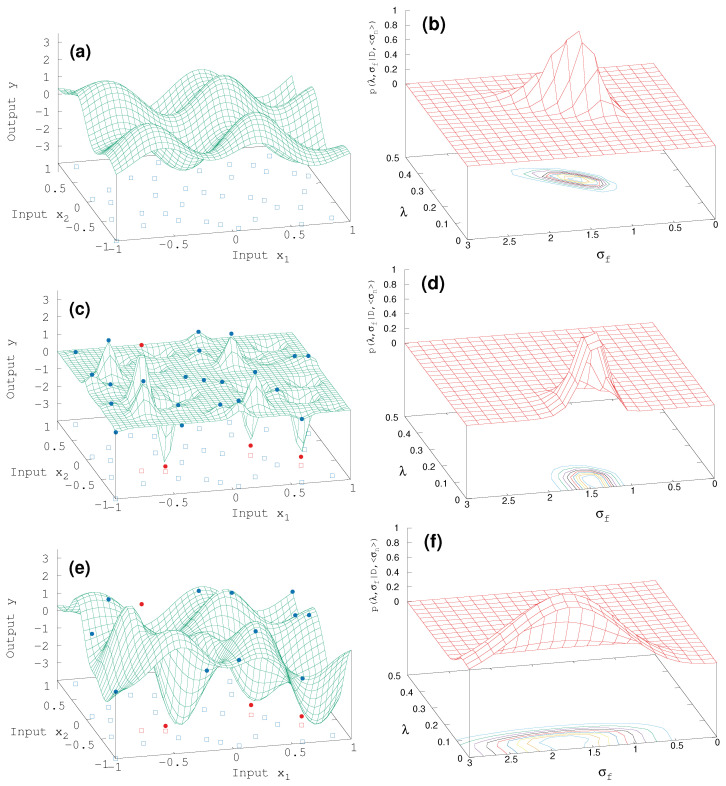
Two-dimensional sin-model data. Surrogate model from Student-t process for first N=40 Sobol data points with added noise of $\sigma_{d}$ = 0.2. (**a**,**b**) GP, no outliers, 〈λ〉 = 0.3 ± 0.04, $\langle \sigma_{f}\rangle$ = 1.3 ± 0.3, $\langle \sigma_{n}\rangle$ = 0.7 ± 0.4; (**c**,**d**) GP, four outliers, 〈λ〉 = 0.06 ± 0.04, $\langle \sigma_{f}\rangle$ = 1.5 ± 0.2, $\langle \sigma_{n}\rangle$ = 1.4 ± 0.9; (**e**,**f**) TP, four outliers, 〈λ〉 = 0.06 ± 0.04, $\langle \sigma_{f}\rangle$ = 1.5 ± 0.2, $\langle \sigma_{n}\rangle$ = 1.4 ± 0.9. Blue dots and their footprints (open squares) in the base are the input data, while the red dots/squares in (**c**,**e**) represent the four outliers. The surrogate surfaces in (**a**,**c**,**e**) are obtained for inserting above expectation values of the hyperparameters into function Equation (Equation 3).

**Figure 4 entropy-25-00685-f004:**
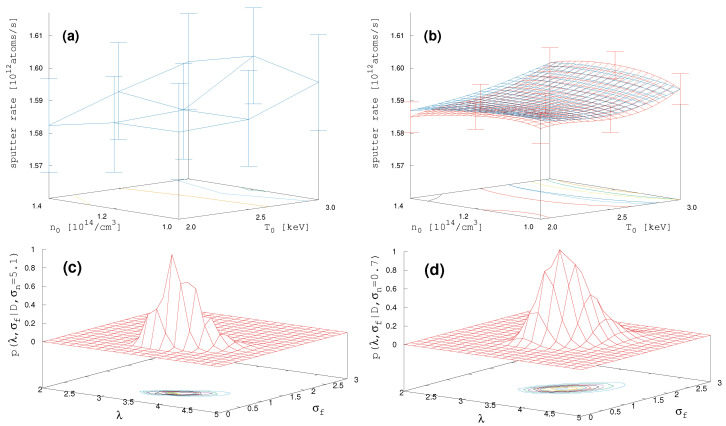
(**a**) EIRENE sputter rate results with errorbars shown in a two-dimensional subspace of parameters {$n_{0}$
$T_{0}$} for max[$T_{\mathrm{ped}}]=8$ keV and min[$n_{\mathrm{ped}}]=0.56\times 10^{14}$/cm^3^. (**b**) Blue mesh: surrogate surface based on initial N=81 EIRENE data. Red mesh: surrogate surface based on a total of N=151 EIRENE data. The surrogate surfaces are obtained for inserting expectation values of the hyperparameters into function Equation (Equation 3). Hyperparameter surfaces of {λ, $\sigma_{f}$} for the results with N=151 data: (**c**) GP; (**d**) TP.

**Figure 5 entropy-25-00685-f005:**
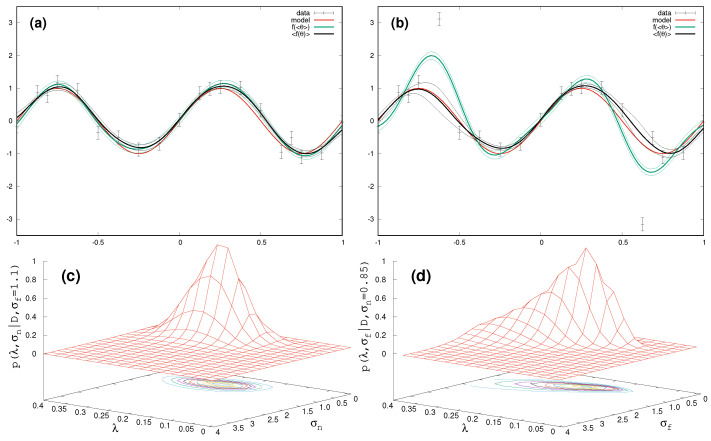
Surrogate model from the Gaussian process with a mixture likelihood for N=20 data points with and without two outliers. (**a**) Normally distributed data following a sin-model. (**b**) Normally distributed data following a sin-model, but the fifth and fifteenth data point were multiplied by a factor of three to simulate outliers. (**c**,**d**) Hyperparameter surfaces for data with outliers, 〈λ〉 = 0.25 ± 0.08, $\langle \sigma_{f}\rangle$ = 1.1 ± 0.5, $\langle \sigma_{n}\rangle$ = 0.85 ± 0.29.

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
