# Peer review of "Outlier-Robust Surrogate Modeling of Ion–Solid Interaction Simulationsâ€"

_entropy, 2023, doi:10.3390/e25040685_

Round 1
Reviewer 1 Report
In this work, the authors revisit Gaussian process regression and Student-T process regression. They compare the posterior distributions of the hyperparameters in the context of outliers, and discuss two variants of the approach where i) a full Bayesian treatment of unknown hyperparameters, i.e. marginalization of the t-process predictions wrt to hparams, and ii) a modification of the contemporary machine learning approach, where the expectation value for hyperparameters is substituted into the model (similar to the MAP value typical in ML). It is demonstrated that substituting MAP estimates is detrimental in this context, and that a Bayesian approach is necessary in order to fully leverage the advantages of the t-process. In another aspect, a Gaussian mixture likelihood is introduced with a GP model to build outlier-robust surrogate models. An application to ion sputter simulation data is presented.
This is a fine, technically solid paper, providing relevant insight into Student-T processes, a generalization of one of the most popular machine learning models.
I only have minor suggestions:
1) On page 5, lines 114-118, the authors state: "For the TP-method (see Figure 1d) the bumps get less pronounced for the expectation values of
the surrogate surface 〈 f (θ)〉 (green line) and disapear completely by just asking for the surrogate surface obtained by inserting the expectation values of the hyperparameters f (〈θ〉) (black line), which clearly follow a sin-function."
Shouldn't it be the other way around? Maybe the model with green line and black line have been switched, or maybe the labels in the figure 1 legend have been mixed
2) In figure 1 a,b,c,d, consider to use dashed lines so it is easier to see the many different lines. The combined surf+contour plots are great.
3) The authors mention "Gray code" several times. Please clarify what this means
4) Consider to indicate in the figure captions or legends more clearly whether expectations values for hyperparameters have been substituted or whether expectation has been done over the process, e.g. in figure 3 it is not clear
5) In Eq. 14, what does "realizations of mixture products" mean?
Typos:
p.6, line 137: "a MAP-solution" instead of "an MAP-solution"
p.9, line 206: "a standard" instead of "an standard"
In the caption of Fig. 3, “while the red dots/squares in (c,f) represent the four outliers” should be “while the red dots/squares in (c,e) represent the four outliers”.
Author Response
Answers to referee 1
1) On page 5, lines 114-118, the authors state: "For the TP-method (see
Figure 1d) the bumps get less pronounced for the expectation values of
the surrogate surface 〈f(θ)〉(green line) and disapear completely by
just asking for the surrogate surface obtained by inserting the
expectation values of the hyperparameters f (〈θ〉) (black line),
which clearly follow a sin-function."
Shouldn't it be the other way around? Maybe the model with green line
and black line have been switched, or maybe the labels in the figure 1
legend have been mixed
Answer:
The referee is completely right. It should read 〈f(θ)〉(black line)
and f (〈θ〉) (green line),
2) In figure 1 a,b,c,d, consider to use dashed lines so it is easier
to see the many different lines. The combined surf+contour plots are
great.
Answer:
We are sorry: we played around with the dashed line style but it even got worse.
After all, the original figures are not so bad and we let them as they are.
3) The authors mention "Gray code" several times. Please clarify what
this means
Answer:
We added an explanation in the introduction.
4) Consider to indicate in the figure captions or legends more clearly
whether expectations values for hyperparameters have been substituted
or whether expectation has been done over the process, e.g. in figure 3
it is not clear
Answer:
We added a sentence in figure 3 and 4.
5) In Eq. 14, what does "realizations of mixture products" mean?
Answer:
We changed "realizations" to "terms"
Typos:
p.6, line 137: "a MAP-solution" instead of "an MAP-solution"
p.9, line 206: "a standard" instead of "an standard"
In the caption of Fig. 3, “while the red dots/squares in (c,f)
represent the four outliers” should be “while the red dots/squares
in (c,e) represent the four outliers”.
Answer:
We thank the referee for his thorough reading!
Reviewer 2 Report
The paper considers a problem of outlier-robust surrogate modelling of ion-solid interaction simulations. The authors compare the performance of Gaussian-process and Student-t process methods using both artificial and real data sets. The authors give a brief overview of related works and describe the proposed method. The paper appears to be well-written and comprehensively referenced.
I have the following comments on the manuscript.
Line 72. Explain the notation < …>. Is this the expected value?
Line 82. Indicate the size of the covariance matrix here.
Formulas (2) and (4). Check the notations var(f_*) and var^{GP}(f_*). Shouldn’t they be the same?
Formulas (3) and (4). Explain \delta.
Line 98. Change N=20 to $N=20$ (see also the captions for Figures 1, 2 and 5). Similarly, change N=40 to $N=40$ in line 142 and the caption for Figure 3. Also, check the font of N in lines 142, 161, 170, 174 and 177, and the caption for Figure 4.
Formula (9) and afterwards. Z_{data} and Z_{outlier} are not appropriate notations for constant weights since Z is traditionally used for the standard normal random variable.
Author Response
Answers to referee 2
Line 72. Explain the notation < …>. Is this the expected value?
Answer:
Explanation inserted: "..., where
the brackets $\langle \ldot \rangle$ indicate an expectation value."
Line 82. Indicate the size of the covariance matrix here.
Answer:
Line 82 mentions only an element of the covariance matrix, so
we clarified this.
Formulas (2) and (4). Check the notations var(f_*) and var^{GP}(f_*). Shouldn’t they be the same?
Answer:
Yes! Thank you, we corrected this.
Formulas (3) and (4). Explain \delta.
Answer:
\Delta is explained in the fourth line of the section (17 lines above) being the
diagonal matrix of the noise.
Line 98. Change N=20 to $N=20$ (see also the captions for Figures 1, 2 and 5). Similarly, change N=40 to $N=40$ in line 142 and the caption for Figure 3. Also, check the font of N in lines 142, 161, 170, 174 and 177, and the caption for Figure 4.
Answer:
We did as requested.
Formula (9) and afterwards. Z_{data} and Z_{outlier} are not appropriate notations for constant weights since Z is traditionally used for the standard normal random variable.
Answer:
We replaced Z by C.